# Rationing of Nursing Care and Its Relationship with Nurse Staffing and Patient Outcomes: The Mediation Effect Tested by Structural Equation Modeling

**DOI:** 10.3390/ijerph16101672

**Published:** 2019-05-14

**Authors:** Xiaowen Zhu, Jing Zheng, Ke Liu, Liming You

**Affiliations:** 1School of Nursing, Jinan University, Guangzhou 510632, China; zhuxiaowen05@163.com; 2School of Nursing, Guangdong Pharmaceutical University, Guangzhou 510310, China; zhengj38@mail.sysu.edu.cn; 3School of Nursing, Sun Yat-sen University, Guangzhou 510080, China; liuke@mail.sysu.edu.cn

**Keywords:** rationing of nursing care, nurse staffing, patient outcomes, structural equation modeling, mediation, missed care, care left undone, unfinished nursing care, underuse of nursing services

## Abstract

*Purpose*: The purpose of this study is to test the mediation effect of rationing of nursing care (RONC) and the relationship this has between nurse staffing and patient outcomes. *Methods*: The analytic sample included 7802 nurse surveys and 5430 patient surveys. Three patient outcome indicators, nurse staffing, RONC, and confounding factors were considered in the model pathways. *Results*: The hypothesized model was shown to be statistically significant. In the model, nurses who were in the units with lower nurse-to-patient ratios reported higher scores on RONC, which meant that an increased level of withheld nursing care or a failure to carry out nursing duties was apparent. Nurses who reported a higher score on RONC, scored poorly on the quality assessment and were more frequently involved in patient adverse events. Nurse staffing influenced quality assessments and patient adverse events through RONC. In units with poorer nurse-reported quality assessments or more frequently patient adverse events, patient-reported dissatisfaction scores were higher. *Conclusions*: The results suggest that a lack of nurse staffing leads to RONC, which leads to poorer patient outcomes. These results are seen when considering the evaluations completed by both nurses and patients. The relationship between staffing numbers and patient outcomes explains the mediating role of RONC.

## 1. Introduction

Rationing of care is a variable that originated from hospital management, and was first introduced to nursing care in 2008. Rationing of nursing care (RONC), refers to necessary nursing tasks that nurses withheld or failed to carry out due to limited time, staffing level, or skill mix [1]. It is an important process factor affecting the quality of patient care. Many nurses feel overworked and indicate problematic conditions in the process of delivering nursing care [2]. They report that they do not have enough time to perform necessary nursing tasks [3]. RONC, in acute general hospitals, is a robust and important predictor of patient outcomes, including patient satisfaction, nurse reported medication errors, bloodstream infections, and pneumonia [4,5]. Due to increasing concern of a lack of nursing human resources and its related impacts, additional studies are needed to develop an in-depth understanding of this issue.

Adequate nurse staffing and organizational support for nursing are both key to improving the quality of patient care [6]. Evidence suggests that when resources are not sufficient to provide all the care needed by their patients, nurses are forced to ration their attention between care activities [1,7,8]. Thus, nursing care omissions occur. Missed nursing care is an approximate part of RONC, which refers to the delay or failure to perform nursing care that results from various reasons [9]. Kalisch, Landstrom, and Williams [10] examined why nursing care was missed, and found that the primary reason was labor resources, including unexpected rise in patient acuity, urgent patient situations, level of staffing, and inadequate number of assistive personnel. A moderate to strong correlation was found between RONC and the nurse work environment, and a significant but low correlation was also show with patient-to-nurse ratios [11]. These two studies both indicated that RONC was related to staffing. A study conducted in the U.S. further found that missed nursing care mediated the relationships between hours per patient day and patient falls [12]. Although nurse staffing levels affect patient fall rates, the level of impact was reduced when care was completed in its entirety. Hospital and nurse leaders in acute-care settings should monitor the level of RONC to decrease the likelihood of negative outcomes for their patients [13]. However, we need to examine more types of patient outcomes to evaluate nursing services.

Nurses in different countries report similar shortcomings in their staffing and the quality of hospital care [14]. In China, it has been demonstrated that inadequate nurse staffing resulted in negative patient outcomes, while better staffing levels was an effective strategy for improving both nurses’ reports of quality of care and patients’ reports of satisfaction [15]. In that multisite study, RONC decreased when the staffing level increased from <0.4 to ≥0.4 in the logistic regression models [15]. However, logistic regression does not allow for causal inferences. Although the importance of rationing as a factor regarding patient safety and quality of care is known [5], studies about RONC in China is still limited [16]. To investigate the role of RONC between established staffing and patient outcomes, more sophisticated statistical methods should be taken into consideration [4].

Although nursing care left undone has been tested as a mediator between work environments, nurses’ workload, and patient safety [17], few researchers have tested a patient outcomes model both considering nurses’ evaluation and patients’ perception. Little is known about how nurses’ evaluation of patient outcomes affects patient satisfaction. The confounding influence of other structural factors including hospital characteristics and unit type should be controlled. Clarifying the roles of RONC in the relationship between nurse staffing and patient outcomes and the association of nurses’ evaluation of patient outcomes and patient satisfaction are of practical and theoretical importance. Such knowledge could help in the development of a strategy to improve quality of care and patient outcomes. The current study explores the interrelationships among the different aspects within nursing work systems using a structural equation model. Based on a literature review and existing knowledge, this study hypothesized that RONC acts as mediator in the relationship between nurse staffing and patient outcomes. To clarify the meaning of the mediation effect, a path diagram is used as a model for depicting a basic causal chain involved in mediation [18].

## 2. Methods

### Study Participants

This analysis used data from a large cross-sectional research project: China Hospital Nursing Workforce Study (CHNWS) [19]. CHNWS recruited nurses and patients using a four-stage sampling design from 181 hospitals from 9 provinces, municipalities, and autonomous regions (PMAs) in mainland China. During the study, the China Nurse Survey was completed voluntarily and anonymously by nurses for direct care. The Hospital Consumer Assessment of Healthcare Providers and Systems (HCAHPS) was completed voluntarily and anonymously by patients that stayed in the same units as the sampled nurses. In this article, 7802 nurse surveys and 5430 patient surveys from medical and surgical units were analyzed.

## 3. Measures

### 3.1. Patient Outcomes

We examined three patient outcome indicators: nurse-reported quality assessments, patient adverse events, and patient-reported dissatisfaction to hospital care. The China Nurse Survey used in this study was confirmed to have good reliability and validity [19]. The China Nurse Survey contains measures of patient adverse events and quality assessments, while the HCAHPS measures patient satisfaction.

Nurse-reported quality assessments were a latent indicator in the structural equation model constituted by three single-item measures: (1) the quality of care usually received by patients on their unit; (2) nurses’ confidence of their patients’ self-care ability on discharge; (3) nurses’ confidence of management being able to act to resolve problems in patient care that they report. These three questions were graded from 1 = excellent/ very confident to 4 = poor/ not at all confident [20]. Higher scores indicate poorer quality.

Nurse-reported patient adverse events were another latent indicator constituted by several items about how frequently adverse events occurred involving their patients, with scores ranging from 1 = never to 7 = every day. Higher scores indicate more frequent patient adverse events. There were 6 patient adverse events in total, including a patient receiving the wrong medication or the incorrect dose, pressure ulcers developing after admission, a patient falling and incurring an injury, the use of physical restraints, healthcare-associated surgical site infections, and urinary tract infection.

Patient-reported dissatisfaction was measured by items from the HCAHPS. To create the patient dissatisfaction summary measure, we used the mean average scores from nine reported measures. The nine reported measures included one of the survey’s global items (“recommend to friends and family”), all six domains as composites (“communication with physicians”, “communication with nurses”, “quality of nursing services”, “pain management”, “communication about medications”, and “adequacy of planning for discharge”), which were constructed from 14 items, and the average of the “cleanliness” and “quietness” of the hospital room environment were stand-alone report items [21,22]. Composite ratings for the domains were calculated by averaging the responses to each individual item within that domain. All included items were 4-point Likert-type items. Higher scores indicate poorer patient satisfaction.

### 3.2. Nurse Staffing

Nurse staffing was calculated as the total number of nurses on all shifts on the unit, divided by patient numbers on the same unit. Total number of nurses on all shifts on the unit was recorded by the research nurse who assisted in collecting data in each unit. Patient numbers on the unit was calculated by the median number of patients who stay on the units reported by nurses. Nurse staffing was divided into four levels: 0 = (nurse-to-patient ratios ≥ 0.6), 1 = (nurse-to-patient ratios 0.5–<0.6), 2 = (nurse-to-patient ratios 0.4–<0.5), 3 = (nurse-to-patient ratios < 0.4) [15].

### 3.3. RONC

RONC was defined as the number of necessary nursing tasks for patients withheld or otherwise not performed due to inadequate time, staffing levels, and/or skill mixes. This variable was measured with “Which of the following activities were necessary but left undone because you lacked the time to complete them in the last working day?” Each of these activities was coded as 1 for a response of “yes” and 0 for a response of “no”. This instrument is a revised version of the Basel Extent of Rationing of Nursing Care (BERNCA-R), which was assessed as having applicable criteria [11]. All the responses of the 12 activities were summed up. A higher score on RONC denoted that more nursing care was withheld or not carried out [1].

### 3.4. Controlling Variables

To adjust for confounding factors, a number of variables that have shown statistical significance in the regression were also been controlled in the model pathways [15]. There are three kinds of controlling variables in this study. First, organization hospital characteristics (2 indicators: Hospital level and location) and unit type. Second, initial education and working time, which have been described as nurse skill mix, were controlled. Third, patient demographics including length of stay, self-rated health status, and educational level from the HCAHPS were also considered as controlling variables.

We chose hospital level and location as hospital characteristics. In China, hospitals are divided into three levels based on their size and academic level. Level 3 hospitals have the largest size and highest academic level. Only Level 2 and Level 3 hospitals were recruited so as to get a high enough nurse and patient sample size for each hospital. Furthermore, because of the variety and regional disparity, location was another considered factor, which was assigned as municipalities, capital cities, and other cities. Medical and surgical units made up the final two unit types that were included in this paper. Initial nursing education, which provides professional skills to nurses, was a key indicator of nurse skill mix. Working time, as a nurse increased their experiences, was also considered. 

### 3.5. Statistical Analysis

Epidata 3.1 (The EpiData Association, Odense, Denmark) was used to input the collected data. Statistical analysis was done with SAS 9.2 system for Windows (SAS Institute Inc., Cary, NC, USA). Basic descriptive analyses were performed for the demographic data of the study subjects and various variables. Logistic regression analyses were applied among the measured variables to examine the relationship between nurse staffing, RONC, and other patient outcomes [15].

Structural equation modeling (SEM) was fitted for a comprehensive assessment of the mediating effect of RONC on the relationship between nurse staffing and patient outcomes. Using the theoretical and empirical literature as a guide, the authors of the present study constructed a model to predict patient outcomes (Figure 1). The pathways were tested in structural equation modeling. The root mean square error of approximation (RMSEA), the comparative fit index (CFI), the goodness of fit index (GFI), the adjusted GFI, and the normal fit index (NFI) were used to examine the overall model fit [23]. The model tested the following hypotheses:When controlling for organizational, nurse, and patient characteristics, the lower the nurse staffing level is, the higher score RONC will be.When controlling for organizational, nurse, and patient characteristics, the lower the nurse staffing level is, the higher score of nurse-reported quality assessments and patient adverse events, and patient-reported dissatisfaction will be.When controlling for organizational, nurse, and patient characteristics, the effects of nurse staffing on patient outcomes were indirect, which were mediated by RONC.When controlling for organizational, nurse, and patient characteristics, the higher the score of nurses-reported quality assessments and patient adverse events lead to higher score of patient-reported dissatisfaction.

## 4. Results

Table 1 describes the subjects’ demographic data. Of the 7802 nurses surveyed in 91 Level 3 hospitals and 90 Level 2 hospitals, the mean age was 29.42 years old (SD = 7.07 years), and average time working as a nurse was 8.73 years (SD = 7.65). Most of the nurses (99.50%) were female, 20.31% had a baccalaureate degree or higher education. However, there were no significant correlations between the nurses’ highest education and patient outcomes [15]. Therefore, nurses’ initial education, which was more symptomatic of nurse skill mix, was chosen in the final structural equation model as well as working time.

As shown in Table 1, of the 5430 patients surveyed, the average age was 54.24 years (SD = 17.65). They had been inpatients for an average of 14.83 days (SD = 18.56). The proportion of male patients was 54.75%. No schooling or primary school (29.29%) and junior middle school (27.85%) were the main educational levels. Almost half of the patients (47.90%) rated their overall health as fair or poor.

RONC has a total of 12 items displayed in Table 2. The mean and standard deviations of the total score of these 12 items were 3.31 and 2.43. Of all the nurses in Level 2 and Level 3 hospitals, 68.2% reported 2 to 4 nursing activities that were rationed. Only 8.17% reported that no nursing activity was rationed. In the results of each single item, more than half of the nurses (63.88% and 55.68%) complained that “comfort/talk with patients” and “teach/counsel patients and family” were left undone. More than 30% of nurses thought that the rationed nursing care also included “adequate patient surveillance” (35.30%), “prepare patients and families for discharge” (35.00%) and “coordinating patient care” (30.25%). Completions of “treatments and procedures” and “administer medications on time” in Chinese nurses were the best two nursing tasks in these 12 items. However, 7.48% and 7.00% of nurses still believed that these tasks were left undone.

Table 3 shows the number and proportion of nurse staffing, nurse-reported quality assessments, patient adverse events, and patient-reported dissatisfaction.

The overall model fit was used to evaluate whether the data fit the theoretical model. According to the overall fit test results, the chi-square of the final structural equation model (Figure 2) fit between the theoretical model and the chi-square data was 3099.22 (*df* = 147, *p* < 0.001). The results showed that the CFI was 0.93, the GFI was 0.99, the adjusted GFI (AGFI) was 0.99, and the NFI was 0.92. These values were all more than 0.90, indicating that the fit was satisfied. In general, the fit indices demonstrated an ideal external quality.

Factor loading of the observed measurement items should be >0.40 to fit a satisfied internal structure of the overall structural model [24]. According to Figure 2, the factor loadings of the three items to the latent variable quality assessments were between 0.63 and 0.96. The factor loadings of the six items to the latent variables patient adverse events were between 0.54 and 0.95. The above analysis demonstrates that the fit of internal structure of latent variables in our final structural equation model reached the standard of statistical significance.

As shown in Figure 2, there was a direct effect of nurse staffing on RONC (*p* < 0.01). In other words, nurses who were in the units with lower nurse-to-patient ratios reported higher scores on RONC, which means more nursing care was withheld or failed to be carried out. At the same time, the direct effects of RONC on nurse-reported quality assessments and nurse-reported patient adverse events also reached statistical significance (*p* < 0.01). These indicate that nurses who reported higher scores on RONC, reported poorer quality assessments and more frequent patient adverse events. According to the standard pathways shown in Figure 2, nurse staffing had only indirect effects on nurse-reported quality assessments and patient adverse events. The influence of nurse staffing on nurse-reported quality assessments and patient adverse events were based on the mediating effects of RONC, and the indirect effects were 0.311 × 0.221 = 0.069, and 0.311 × 0.122 = 0.038 (*p* < 0.01). This suggests that nurse staffing influenced quality assessments and patient adverse events through RONC. In other words, nurse staffing had only indirect effect on nurse-reported patient outcomes. Nurses who were in the units with lower nurse-to-patient ratios reported poorer quality assessments and more frequently patient adverse events.

The direct effects of nurse-reported quality assessments and patient adverse events on patient-reported dissatisfaction also reached statistical significance (*p* < 0.01), controlling for patient demographics (length of stay, self-rated health status, and patient’s educational level) and other confounding factors. These indicate that in units with poorer nurse-reported quality assessments or more frequently patient adverse events, there was poorer patient satisfaction. Nurse staffing also had an indirect effect on patient-reported dissatisfaction. The indirect effect was 0.311 × 0.221 × 0.092 + 0.311 × 0.122 × 0.028 = 0.007 (*p* < 0.01). In other words, in the units with lower staffing, nurses reported higher RONC, and thus reported poorer patient quality. It follows that this led to lower patient satisfaction.

## 5. Discussion

The findings of this study support our proposed hypothesized model. The results shown in this structural equation model provide strong support for the close association between nurse staffing, RONC, and patient outcomes. In regard to hospital outcomes, nurse staffing level in the unit has been shown to be a leading factor. This finding is in agreement with other studies [25]. However, most units’ nursing staff level in this study was under 0.4. If we assume a unit serves 40 patients, that means that in more than half the units, less than 16 clinical nurses are available, including both day shift and night shifts. This lack of nursing resources hinders the ability of the hospital to provide a high level of hospital care.

Our study has shown that insufficient human resource leads to RONC. After controlling for hospital characteristics, unit type, and nurse skill mix, RONC increased while nurse staffing decreased, which confirmed our first hypothesis. This result was in line with the international data from the RN4CAST study, which suggested better unit level staff resource adequacy was consistently significantly associated with lower rationing levels [26]. According to Donabedian, good structure increased the likelihood of good process [27]. Nurse staffing as a structured factor affected the process factor, which was RONC. In this study, RONC was shown to be a routine occurrence existence in clinical nurses’ work. On average, there were more than three tasks left undone. Mental health nursing and health education were the most frequently missed tasks. Nearly two thirds of nurses reported comfort or talk with patients were necessary but left undone because lack of time; and one fifth of nurse had no time to care about patients’ pain. Health education was not much better. Half of the nurses rationed their time in regard to teaching or counselling patients and family, and one third of nurses missed preparation of patients and families for discharge.

RONC also involves concerns in other studies. Nurses who worked in intensive care units (ICUs) with mixed work environments were inclined to report better rationing than nurses who worked in ICUs with poor work environments [16]. Kalisch, in a qualitative study conducted in 2006, found that there were nine common missed nursing tasks: Ambulation, turning, feeding, patient teaching, discharge planning, emotional support, hygiene, intake and output documentation, and surveillance [9]. After that, Kalisch et al. reported, in a quantitative study, that most missed nursing tasks were in emergency hospitals, where 73% of the nurses reported interventions of individual needs and basic care were missed, 71% of the nurses’ missed planning, and 44% of nurses’ missed assessment [10]. Similarly, nutritional assessments and records are often missing [28,29].

In contrast, medication and treatment, as left undone tasks, were seen much less frequently than others (both proportions were about 7%). It is likely that these tasks are what we called “rigid targets” in Chinese hospitals which must be completed first. Schubert et al. found that in Switzerland, tasks related to medical, technical, and therapeutic treatments were less frequently rationed than those in the areas of caring and support [26]. In addition to the time allocation in different nursing tasks, some nurses complained that they always or nearly always wasted time locating or borrowing equipment from other wards. In a study conducted in Guangdong province in south China in 2014, together with nurses’ work environment, non-professional tasks carried by nurses were confirmed to influence good quality nursing care [16]. RONC and these process components should receive an increasing level of attention in providing nursing care.

When controlling for all the confounding factors including organizational, nurse, and patient factors, RONC mediated the effects, which confirmed our second hypothesis. The mediation of RONC helps to explain how the structural component of nurse staffing affected patient outcomes. Studies, such as those recently adopted in England, also found that patients’ perceptions of care were significantly eroded by with increases in missed nursing care [30]. One of this paper’s tasks was to study patient outcomes—both considering nurses’ evaluation and patients’ perception. In the two latent models of “nurse-reposted quality assessments” and “patient adverse events”, factor loading estimates for all regression paths were > 0.40 (*p* < 0.01), meeting the traditional cutoff value [31]. In the latent model of “nurse-reported quality assessments”, the factor loading of the quality of care usually received by patients on their unit was the highest (0.962), indicating that this index was the main measurement index of nurse-reported quality assessments. In the measurement model of “patient negative event”, the factor loading of drug administration error was the highest (0.953), indicating that this index was the main measurement index of patient negative events.

When controlling for hospital characteristics, unit type, nurse skill mix, and patient demographics, nurses’ evaluation of quality assessments and patient adverse events had positive effects on patient-reported dissatisfaction, which confirmed our final hypothesis. The higher score of nurses-reported quality assessments and patient adverse events led to a higher score of patient-reported dissatisfaction. This result showed us that nurses and patients could both represent the outcomes of nursing care. It was also shown that nearly half of the patients would definitely not recommend their hospital to friends and family. This indicated that a great many patients were reluctant to recommend their experiences, which implies that our clinical practice still has room for improvement. The quality and safety of nursing care, especially whether there were adverse events aimed at patients would influence their satisfaction.

The study had several limitations. First, it was conducted with a sample of nurses and patients who stayed at medical and surgical units in nine provinces, municipalities, and PMAs in mainland China, which may limit its generalizability to other cultures and regions. Second, all data that were used in this study were self-reported. Third, the study was cross-sectional; future longitudinal studies could address how nurse staffing changes over time and how RONC develops as a factor in the model. Fourth, the explanatory power of independent variables on dependent variables in the final structural equation model is low. The model could be further expanded in complexity and explanatory power by introducing additional relevant nursing process variables that have not been examined among clinical settings. In future studies, data from different sources would not only increase the generalizability of the model, but will also help to identify additional meaningful factors that could contribute to explaining patient outcomes. Despite these limitations, our results yield meaningful information about nurse staffing, RONC, and related patient outcomes.

## 6. Conclusions and Implications

In conclusion, this study explored the interrelationships among nurse staffing, RONC, and patient outcomes. The results suggest that RONC plays a mediating role in the relationship between staffing and patient outcomes after controlling for hospital and patient factors.

Regarding possible implications for clinical practice and for managers, the identified RONC influencing factors can determine starting points for an interventional program for nurse managers to promote good patient outcomes. With this in mind, one obvious starting point would be to improve nursing resources. As this approach could logically be expected to lead to reduced RONC levels, which will then reduce the risk and expense of negative patient outcomes. Targeting RONC may be another more effective intervention. Thus, an interventional program for nurse managers designed to relieve RONC may be useful to improve patient outcomes in China.

## Figures and Tables

**Figure 1 ijerph-16-01672-f001:**
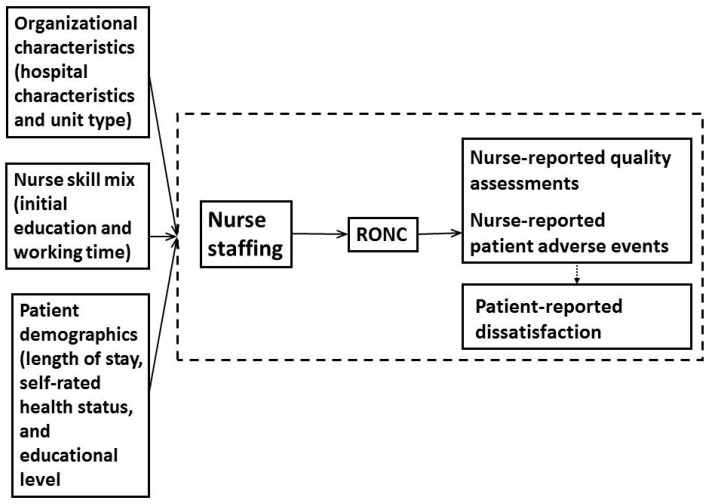
Hypothesized relationships among organizational characteristics, nurse skill mix, patient demographics, nurse staffing, rationing of nursing care (RONC), and patient outcomes.

**Figure 2 ijerph-16-01672-f002:**
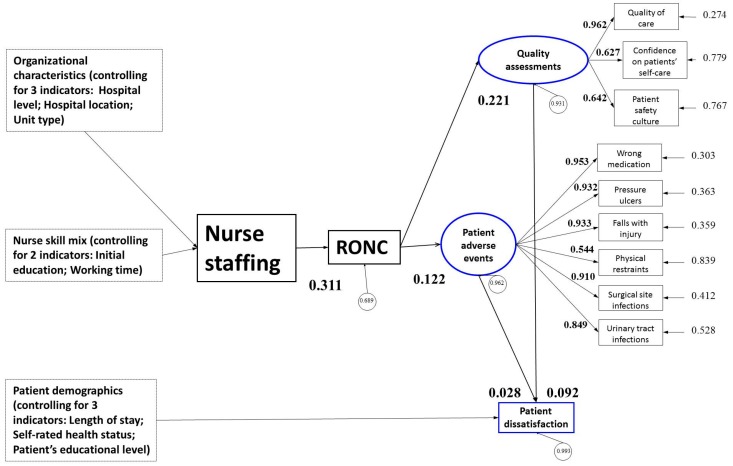
Final structural equation model of nurse staffing and patient outcomes among Chinese hospitals with estimated standardized path coefficients and correlations. Path coefficients were significant (*p* < 0.01). Otherwise coefficients are not shown for clarity.

**Table 1 ijerph-16-01672-t001:** Demographics of nurse (*n* = 7802) and Patient (*n* = 5430).

Nurse	*n* (%)	Patient	*n* (%)
Age (year) (Mean (SD))	29.42 (7.07)	Age (year) (Mean (SD))	54.24 (17.65)
Working years as a nurse (Mean (SD))	8.73 (7.65)	Length of stay (Mean (SD))	14.83 (18.56)
Gender		Gender	
Female	7612 (99.50)	Female	2422 (45.25)
Male	38 (0.50)	Male	2930 (54.75)
Initial education		Education	
Secondary diploma	4961 (64.29)	No schooling or primary	1569 (29.29)
Advanced diploma	2374 (30.76)	Junior middle school	1492 (27.85)
BSN degree	382 (4.95)	Senior middle school	1231 (22.98)
Highest education		College or higher	1065 (19.88)
Secondary diploma	1418 (18.41)	Self-rated health	
Advanced diploma	4720 (61.28)	Good to excellent	2787 (52.10)
BSN degree or higher	1564 (20.31)	Fair or poor	2562 (47.90)

**Table 2 ijerph-16-01672-t002:** Description of RONC (*n* = 7688).

Item	*n*	%
RONC (Mean (SD))	3.31	2.43
Response of “yes” of the 12 items in RONC ^a^		
Comfort/talk with patients	4911	63.88
Teach/counsel patients and family	4281	55.68
Adequate patient surveillance	2713	35.30
Prepare patients and families for discharge	2691	35.00
Coordinating patient care	2321	30.25
Develop or update nursing care pans	1806	23.49
Skin care	1682	21.88
Pain management	1629	21.19
Adequately document nursing care	1334	17.35
Oral hygiene	936	12.17
Treatments and procedures	575	7.48
Administer medications on time	538	7.00

^a^ The missing values were not calculated in frequency and percentage.

**Table 3 ijerph-16-01672-t003:** Description of nurse staffing, nurse-reported quality assessments, nurse-reported patient adverse events and patient-reported dissatisfaction ^a^.

Variables	*n* (%)
Nurse staffing (*n* = 599)	
0.11–<0.40	413 (68.95)
0.40–<0.5	94 (15.69)
0.50–<0.6	46 (7.68)
0.60–1.40	46 (7.68)
Nurse-reported quality assessments	
Quality of care as fair or poor (*n* = 7735)	2351 (30.39)
Not confident about patients’ self-care ability on discharge (*n* = 7736)	3384 (43.74)
Patient safety culture as poor or failing (*n* = 7493)	261 (3.48)
Nurse-reported patient adverse events as “once a month or less” or more frequently	
Wrong medication (*n* = 7684)	337 (4.39)
Pressure ulcers (*n* = 7661)	319 (4.16)
Falls with injury (*n* = 7624)	175 (2.30)
Physical restraints (*n* = 7614)	1274 (16.73)
Surgical site infections (*n* = 7516)	504 (6.71)
Urinary tract infections (*n* = 7563)	655 (8.66)
Patient-reported dissatisfaction	
Not definitely recommend this hospital to friends and family (“probably yes”, “probably no” and “definitely no”) (*n* = 5371)	2513 (46.79)

^a^ Sample size varied because of different numbers of units, nurses and patients and also missing data.

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
