# Peer review of "Rationing of Nursing Care and Its Relationship with Nurse Staffing and Patient Outcomes: The Mediation Effect Tested by Structural Equation Modeling"

_ijerph, 2019, doi:10.3390/ijerph16101672_

Round 1
Reviewer 1 Report
Thank you for the opportunity to review this manuscript. The topic is highly relevant and very timely. Overall this was a very well executed study and the manuscript is well done. I have minor suggestions. First, I suggest adding keywords to link this work with other similar work. There are multiple terms used for the same phenomenon and it is helpful to link them for readers. I suggest adding missed care, care left undone, unfinished nursing care, and underuse of nursing services.
The English is quite good in this manuscript but a few grammatical issues remain. Mostly relating to verb structures. For example, page 2/11 line 72 - few researchers have is the correct grammar. Also, on page 2/11 line 57, you indicate "patient outcomes should include more variables." I think you mean that either we need more variables to explain patient outcomes or that we need to examine more types of patient outcomes to evaluate nursing services. So, a close editing for remaining grammar would be helpful.
Study limitations were adequately captured. Perhaps a comparison of results to the RN4Cast findings would also be helpful. How does China compare to those countries?? The RN4cast is a major body of work related to RONC and really should be mentioned.
Author Response
Response to Reviewer 1 Comments
Point 1: Thank you for the opportunity to review this manuscript. The topic is highly relevant and very timely. Overall this was a very well executed study and the manuscript is well done. I have minor suggestions. First, I suggest adding keywords to link this work with other similar work. There are multiple terms used for the same phenomenon and it is helpful to link them for readers. I suggest adding missed care, care left undone, unfinished nursing care, and underuse of nursing services.
Response 1: More key words including missed care, care left undone, unfinished nursing care, and underuse of nursing services are added.
Point 2: The English is quite good in this manuscript but a few grammatical issues remain. Mostly relating to verb structures. For example, page 2/11 line 72 - few researchers have is the correct grammar. Also, on page 2/11 line 57, you indicate "patient outcomes should include more variables." I think you mean that either we need more variables to explain patient outcomes or that we need to examine more types of patient outcomes to evaluate nursing services. So, a close editing for remaining grammar would be helpful.
Response 2: All the words and grammar are checked again and some mistakes are corrected. The mistakes are as follow. “Patient outcomes should include more variables to evaluate nursing care” is changed to “we need to examine more types of patient outcomes to evaluate nursing services” in line 55, p.2. “Responses for the 12 activities” is changed to “responses of the 12 activities” in line 134, p.3. “Other ward” is changed to “other wards” in line 288, p.9. “Factor load” is changed to “factor loading” in line 303, p.9. “Negative event” is changed to “negative events” in line 304, p.9. “Nurse staffing, RONC, related to patient outcomes” is changed to “nurse staffing, RONC, and related patient outcomes” in line 327, p.10.
Point 3: Study limitations were adequately captured. Perhaps a comparison of results to the RN4Cast findings would also be helpful. How does China compare to those countries?? The RN4cast is a major body of work related to RONC and really should be mentioned.
Response 3: Study results in other countries such as England and Switzerland sponsored by RN4CAST are added in discussion in line 261, P.8, line 284, P.9, and line 294, P.9.

Reviewer 2 Report
The manuscript raises a topic of great interest, such as to test the mediation effect of rationing of nursing care (RONC) in the relationship between nurse staffing and patient outcomes. In general, the manuscript is understandable and complete.
Aspects of improvement:
In the methodology it would be interesting to explain more extensively the concept to RONC. Also, in the results, it would be interesting to know in more detail the ratios according to the type of hospital service. The discussion should comment more results. The results of tables 2 and 3 are of great interest and are discussed little, for example in Table 3, it is indicated: "Not definitely recommend this hospital to friends and family (n=5,371) 2,513 (46.79)", this should be discussed in the discussion from a more critical point of view, the possible reasons, etc.
Finally,It would also be of interest to explain more specifically the implications of these results for improving clinical practice and for managers. For example, what objectives and characteristics would an interventional program for nurse managers have designed to improve patient outcomes?
Author Response
Response to Reviewer 2 Comments
Point 1: The manuscript raises a topic of great interest, such as to test the mediation effect of rationing of nursing care (RONC) in the relationship between nurse staffing and patient outcomes. In general, the manuscript is understandable and complete.
Aspects of improvement:
In the methodology it would be interesting to explain more extensively the concept to RONC.
Response 1: Description of RONC, especially its application and validation are now strengthened in measures.
Point 2: Also, in the results, it would be interesting to know in more detail the ratios according to the type of hospital service.
Response 2: Specific level of recruited hospitals now is added in line 180, p.5. as “in 91 Level 3 hospitals and 90 Level 2 hospitals”.
Point 3: The discussion should comment more results. The results of tables 2 and 3 are of great interest and are discussed little, for example in Table 3, it is indicated: "Not definitely recommend this hospital to friends and family (n=5,371) 2,513 (46.79)", this should be discussed in the discussion from a more critical point of view, the possible reasons, etc.
Response 3: Discussion of the result of patient satisfaction in table 3 is now integrated in line 310, p.9 as “It was also shown that nearly half of the patients were not definitely recommend their hospital to friends and family in the results. This indicated that a great many patients were hesitate when they consider recommendation, which imply to us our clinical practice still had room for improvement. The possible reason lied in healthcare provides.”
Point 4: Finally,It would also be of interest to explain more specifically the implications of these results for improving clinical practice and for managers. For example, what objectives and characteristics would an interventional program for nurse managers have designed to improve patient outcomes?
Response 4: The specific intervention implication for clinical practice and managers are written more explicit as “Regarding possible implications for clinical practice and for managers, the identified RONC influencing factors can determine starting points for an interventional program for nurse managers to promote good patient outcomes. With this in mind, one obvious starting point would be to improve nursing resources. As this approach could logically be expected to lead to reduced RONC levels, which then reduced the risk and expense of negative patient outcomes. To target RONC may be another more effective intervention. Thus, an interventional program for nurse managers designed to relieve RONC may be useful to improve patient outcomes among China.”